# Review on Multispectral Photoacoustic Analysis of Cancer: Thyroid and Breast

**DOI:** 10.3390/metabo12050382

**Published:** 2022-04-22

**Authors:** Seongyi Han, Haeni Lee, Chulhong Kim, Jeesu Kim

**Affiliations:** 1Departments of Cogno-Mechatronics Engineering and Optics & Mechatronics Engineering, Pusan National University, Busan 46241, Korea; hsi1748@pusan.ac.kr (S.H.); haenilee@pusan.ac.kr (H.L.); 2Departments of Convergence IT Engineering, Mechanical Engineering, and Electrical Engineering, School of Interdisciplinary Bioscience and Bioengineering, Medical Device Innovation Center, Pohang University of Science and Technology (POSTECH), Pohang 37673, Korea; chulhong@postech.edu

**Keywords:** photoacoustic imaging, thyroid cancer, breast cancer, multispectral analysis, clinical imaging

## Abstract

In recent decades, photoacoustic imaging has been used widely in biomedical research, providing molecular and functional information from biological tissues in vivo. In addition to being used for research in small animals, photoacoustic imaging has also been utilized for in vivo human studies, achieving a multispectral photoacoustic response in deep tissue. There have been several clinical trials for screening cancer patients by analyzing multispectral responses, which in turn provide metabolomic information about the underlying biological tissues. This review summarizes the methods and results of clinical photoacoustic trials available in the literature to date to classify cancerous tissues, specifically of the thyroid and breast. From the review, we can conclude that a great potential exists for photoacoustic imaging to be used as a complementary modality to improve diagnostic accuracy for suspicious tumors, thus significantly benefitting patients’ healthcare.

## 1. Introduction

In biomedical studies, the characterization of molecular and functional information about the underlying tissue can significantly improve our intuition in analyzing the morphology, treatment efficacy, and metabolites of target tissues. Among many biomedical imaging techniques, optical imaging methods have been widely applied for small animal studies due to their cost-efficiency, ease of implementation, non-ionizing radiation, and real-time imaging capability [1]. More importantly, optical imaging techniques can also provide molecular and functional information by tuning the wavelength of the light source. While advantageous, the strong optical diffusion of the pure optical imaging techniques in biological tissues leads to a reduced penetration depth, thus limiting clinical translation.

Photoacoustic imaging (PAI), one of the branches of optical imaging, provides the added advantage of increased imaging depth [2]. Compared to other optical imaging techniques, PAI inherits ultrasound imaging characteristics (USI), which increases its ability to visualize structural information in deep tissue. The signal generation in PAI is based on the photoacoustic (PA) effect, which is energy transduction from light to ultrasound (US) [3]. In brief, PA images can be achieved through the following procedure: (i) pulse laser illumination, (ii) light absorption by chromophores, (iii) momentary heat generation, (iv) acoustic wave (i.e., PA wave) generation through thermoelastic expansion, (v) signal detection by US transducer, and (vi) image generation. The resulting PA images, formed from the acoustic wave, include the optical absorption characteristics of the underlying biological tissue. Thus, PA images can provide molecular and functional information with a good ultrasound resolution in deep tissue [4,5]. In addition to the endogenous chromophores, contrast-enhanced PAI [6,7] has been studied by developing exogenous contrast agents including organic dyes [8,9,10,11,12] and inorganic nanoparticles [13,14,15,16,17,18]. Recently, contrast agents that absorb light in the second near-infrared region (NIR-II, 1000–1350 nm) have been investigated. In NIR-II, a greater imaging depth can be achieved with reduced tissue scattering and background noise compared to the first near-infrared region (NIR-I, 650–950 nm), which is mainly used for contrast-enhanced PAI [19,20,21,22,23,24,25].

One unique advantage of PAI is its scalable resolution and large imaging depth for the target region [26]. Since laser excitation can be tightly focused in shallow areas, high-resolution PA images can be achieved within the optical diffusion limit (~1 mm under the skin) [27,28]. High-resolution PAI has been used for visualizing the hemodynamics of the brain, ear, and eye of mice in vivo [29,30]. Since the light is diffused beyond the optical diffusion limit for deeper imaging depths, the US transducers determine the resolution at a greater depth [31,32,33,34].

For clinical research, PAI platforms have been developed and applied. Among them, the Vintage series (Verasonics, Kirkland, WA, USA), which equips the most-advanced programmable platform for designing user-defined operation sequence, is widely used [35,36]. The VevoLAZR series (FujiFilm VisualSonics, Toronto, ON, Canada) is another widespread commercial system for PAI research [37,38]. The main advantage of this system is a user-friendly interface, with real-time imaging and spectral analysis capability. Its high-frequency US transducer can provide high-resolution images, but it also limits the application area to a shallow area which is not favorable for clinical translation. The MSOT Acuity series (iThera Medical, Munich, Germany) has also been applied in clinical trials with multispectral PA analyses [39,40]. Its arc-shaped array can provide volumetric images, but its relatively small field of view is not suitable for general clinical applications that require a large imaging area. An FDA-cleared US machine (EC-12R, Alpinion Medical Systems, Anyang, Korea) has also been used to develop a clinical PAI system [41,42]. From the programmable platform in the US machine, the user can design their own operation sequence for their specific application.

Various geometry of multi-element array transducers has been developed and used for the clinical translation of PAI [43,44,45,46,47]. In typical clinical PAI systems, both US and PA images are acquired by controlling the data acquisition sequence [48]. The dual-modal images complement each other by visualizing molecular and functional information in the PAI and specific morphologies in the USI [49]. Dual-modal PA and US imaging (PAUSI) has been used for clinical investigation in humans [50,51]. The multispectral PA responses provide metabolomic information about the biological tissues, thus indirectly providing valuable information about the cancerous tissues [52,53].

Among the various clinical trials for screening cancerous tissues that have been studied using multispectral PAI, this article focuses on reviewing the recent results from clinical trials investigating thyroid and breast cancers. The system configuration details, patient recruiting method, and classification results are summarized. A detailed analysis of thyroid and breast cancers provides insights into the future directions of PAI-based cancer diagnoses.

## 2. Principles of Multispectral Photoacoustic Analysis

Similar to other optical imaging techniques, PA investigates the molecular composition of chromophores by applying spectral unmixing techniques to multispectral data [54]. From the images acquired with various excitation wavelengths, the spectral responses from different chromophores are delineated by mathematical calculations. The typical application of the spectral unmixing technique in PAI is in calculating hemoglobin oxygen saturation (sO_2_) levels in blood vessels [55,56,57]. The sO_2_ level in tumors indirectly represents metabolomic information about the tissue. Thus, PA-guided tumor assessment typically measures the sO_2_ levels to differentiate benign from malignant tumors [58,59,60].

The typical unmixing method for multispectral PA data is linear unmixing, which can be simplified as a matrix multiplication as shown below:(1)$$\left[\begin{array}{ccc} \begin{array}{cc} V\left(P_{1},\lambda_{1}\right) & V\left(P_{2},\lambda_{1}\right)\\ V\left(P_{1},\lambda_{2}\right) & V\left(P_{2},\lambda_{2}\right) \end{array} & \cdots & \begin{array}{c} V\left(P_{K},\lambda_{1}\right)\\ V\left(P_{K},\lambda_{2}\right) \end{array}\\ \vdots & \ddots & \vdots \\ V\left(P_{1},\lambda_{N}\right)\;\;\;V\left(P_{2},\lambda_{N}\right) & \cdots & V\left(P_{K},\lambda_{N}\right) \end{array}\right]=\left[\begin{array}{cc} \mu_{Hbo}\left(\lambda_{1}\right) & \mu_{HbR}\left(\lambda_{1}\right)\\ \mu_{Hbo}\left(\lambda_{2}\right) & \mu_{HbR}\left(\lambda_{2}\right)\\ \vdots & \vdots \\ \mu_{HbO}\left(\lambda_{N}\right) & \mu_{HbR}\left(\lambda_{N}\right) \end{array}\right]\left[\begin{array}{cccc} C_{Hbo}\left(P_{1}\right) & C_{Hbo}\left(P_{2}\right) & \cdots & C_{Hbo}\left(P_{K}\right)\\ C_{HbR}\left(P_{1}\right) & C_{HbR}\left(P_{2}\right) & \cdots & C_{HbR}\left(P_{K}\right) \end{array}\right]$$
(2)V=MC
where $P_{i}\;\left(i=1,\;2,\;\cdots ,\;K\right)\;$ is the i-th pixel in images and $\lambda_{j}\;\left(j=1,\;2,\;\cdots ,\;N\right)$ is the j-th excitation wavelength, $V\left(P_{i},\;\lambda_{j}\right)$ is the measured value, $\mu_{HbO}\left(\lambda_{j}\right)$ and $\mu_{HbR}\left(\lambda_{j}\right)$ are optical absorption coefficients, and $C_{HbO}\left(P_{i}\right)$ and $C_{HbR}\left(P_{i}\right)$ are concentrations of oxy-hemoglobin (HbO) and deoxy-hemoglobin (HbR), respectively. The concentrations of hemoglobins are calculated by taking the pseudo-inverse as shown below.
(3)$$C={\left(M^{T}M\right)}^{-1}M^{T}V$$

Consequently, the concentration of the total hemoglobin (HbT) and the sO_2_ level in each pixel are derived using the following equations.
(4)$$C_{HbT}\left(P_{i}\right)=C_{HbO}\left(P_{i}\right)+C_{HbR}\left(P_{i}\right)$$
(5)$$C_{sO_{2}}\left(P_{i}\right)=\frac{C_{HbO}\left(P_{i}\right)}{C_{HbT}\left(P_{i}\right)}$$

In addition to the linear unmixing technique, the model-based unmixing [61] and the deep-learning technique [62] have also been used for multispectral PA analysis.

## 3. Multispectral Photoacoustic Analysis of Thyroid Gland

Thyroid cancer is one of the most common cancers, with an increasing global incidence rate in men and women [63,64,65]. The gold standard for diagnosing thyroid nodules is fine-needle aspiration biopsy (FNAB) [66]. The triaging for FNAB of the nodule is determined by the characteristics of nodules in USI [67,68,69]. Although the sensitivity of US-guided triaging is greater than 90%, the lack of functional metabolomics results in a low specificity of 20–50% [70]. The high false-positive rate leads to unnecessary FNAB, which results in the over-diagnosing of the tumor. Thus, clinical trials have been conducted to enhance the accuracy of triaging thyroid nodules using PAI due to its molecular and functional imaging capability.

Dogra et al. analyzed 88 resected tissues (13 malignant nodules, 30 benign nodules, 13 colloid accumulations, and 32 normal tissues) from 50 patients (11 malignant and 39 benign) [71]. Four different wavelengths (760, 850, 930, and 970 nm) were used for the spectral unmixing of HbO, HbR, lipid, and water components from multispectral PA data. Statistically significant differences were found in HbO and HbR between malignant and other types of tissues (Figure 1a). In particular, HbR components were significantly different between malignant and normal tissue, with a *p*-value of 0.003 in the student *t*-test. The results showed the promising feasibility of PA-guided classification with a sensitivity of 69.2% and a specificity of 96.9%, but this study was limited to ex vivo environments only. Thus, for clinical translation, further in vivo validation is needed.

Dima et al. demonstrated the in vivo imaging capability of their PA and US system for the human thyroid [72]. They recruited two healthy volunteers to acquire PA images with a single excitation wavelength of 800 nm (Figure 1b). US Doppler images were also acquired in the same region to verify the blood vessel’s position (yellow box in Figure 1b). By comparing the PA images with the US Doppler images, surrounding blood vessels extending from the isthmus and carotid artery to the anterior of the thyroid gland were identified. The results showed the feasibility of in vivo PAI using the arc array US transducer by confirming the matched positions of blood vessels (white arrows in Figure 1b). However, the spectral analysis of cancerous nodules was not available in this study. Yang et al. compared in vivo PA images between papillary thyroid cancer (PTC) patients and healthy volunteers (Figure 1c) [73]. Although they achieved PA responses from cancerous nodules, the number of patients (10 PTC and 3 normal) included in the study was insufficient for statistical analysis. In addition, multispectral analysis was also not available in this study because they used a single excitation wavelength of 1064 nm.

Roll et al. presented multispectral PA analyses for differentiating tissue disorders in the thyroid gland [74]. The composition of HbO, HbR, fat, and water were spectrally unmixed from the in vivo PA images of the enrolled patients (6 Graves’ disease, 3 malignant, 13 benign, and 8 healthy), obtained using eight excitation wavelengths (700, 730, 760, 800, 850, 900, 920, and 950 nm). The sO_2_ levels of the thyroid were also visualized and investigated (Figure 2a). The contours of the thyroid glands were determined by the corresponding US images. Statistical analyses demonstrated significant differences between diseased and normal thyroid tissues (Figure 2b). This study was also unable to calculate the classification accuracy due to a small number of samples, thus limiting the applicability of the results.

Recently, Kim et al. presented a multispectral PA analysis with a statistically sufficient number of samples (23 PTC and 29 benign), the largest number of patients in a clinical thyroid study to date [75]. They achieved multispectral PAI using five wavelengths (700, 756, 796, 866, and 900 nm). The corresponding US data were also acquired simultaneously for delineating the boundary of nodules. Similar to the previous studies, the sO_2_ levels in nodules were acquired through the spectral unmixing of HbO and HbR (Figure 2c). Three parameters were quantified and used for training the decision function (Figure 2d): (i) PA spectral gradient: the slope of a first-order polynomial fitted line to the average value of the top 50% of PA signals within the nodule boundary at each wavelength; (ii) relative sO_2_ level: the average value of the top 50% sO_2_ values within the nodule; (iii) skewed angle of sO_2_ distribution: the skewed angle of the Gaussian-fitted distribution for the top 50% of sO_2_ values within the nodule. With the values of the three parameters scattered in a 3D plane, a support vector machine was trained to determine the 3D decision boundary, which showed a good classification accuracy with a sensitivity of 78% and a specificity of 93%. The classification accuracy was further enhanced using a novel scoring method (ATAP score), which combined a conventional USI-based scoring method (i.e., ATA guideline score) and the photoacoustic probability of malignancy. The novel scoring method improved the sensitivity to 83% and the specificity to 93%. Thus, the results showed a great potential for enhancing the triaging accuracy of thyroid nodules using a multiparametric analysis of multispectral PA data as a complementary method to the conventional triaging method.

While PA analyses of thyroid nodules have been conducted by various groups worldwide (Table 1), the validation of multispectral PA analysis is still at the initial stage of evaluation. Further studies are required to address the following issues for successful clinical translation. (i) Larger number of patients are needed to enhance the reliability of this technique. (ii) In addition to PTC, the classification of other types of thyroid cancers such as follicular, medullary, and anaplastic thyroid cancers would expand the application area. (iii) Quantitative analyses of PA responses in skin color are needed. (iv) System improvement with a deeper imaging depth, faster frame rate, and smaller size would enhance the image quality for multispectral analyses.

## 4. Multispectral Photoacoustic Analysis of Breast

Breast cancer is the most common cancer for women worldwide [76,77], with more than 110,000 new cases estimated to be diagnosed in the United States alone in 2022 [78]. Similar to other cancers, early diagnosis of breast cancer increases the survival rates of affected patients [79]. Diagnostic imaging modalities such as X-ray digital mammography, magnetic resonance imaging (MRI), and USI are currently used for screening breast cancers. While X-ray mammography is the most commonly used screening device and has been found to reduce mortality, its sensitivity and specificity are low, especially in dense breast patients and small tumors [80,81,82]. MRI has a high sensitivity in detecting cancer but its specificity is low, resulting in unnecessary biopsies. In addition, MRI is cost-prohibitive and is not available for routine examination [83]. The low-cost, commonly available USI can overcome the disadvantages of mammography, especially in dense breasts, but the lack of functional information degrades its diagnostic accuracy [84]. Thus, there have been efforts to use PAI to increase diagnostic accuracy in breast cancer by providing molecular and functional information with multispectral analysis. By combining this with structural information (shape of nodules) obtained by USI, the PAI showed a great potential to distinguish benign from malignant tumors in vivo.

The feasibility of PAI for in vivo human breast has been evaluated by several researchers [85,86,87]. Hemispherical array transducers have typically been used for the 3D visualization of the breast. Kruger et al. successfully visualized 3D volumes of breasts from four healthy volunteers (Figure 3a) [88]. The resulting image clearly showed the blood vessels, with a sufficient contrast-to-noise ratio (CNR) of ~200 and an imaging depth of 30 mm. However, since they used a single-wavelength (756 nm) laser source, the results lacked functional information. Schoustra et al. acquired the 3D PA images of two healthy breasts using arc-shaped detector arrays and a laser with two wavelengths (755 and 1064 nm) [89]. Although they used multiple wavelengths, the study’s goal was focused on penetration depth and optical absorption (Figure 3b).

Lin et al. acquired tomographic PA images of the breast using a full-ring array transducer with a single wavelength of 1064 nm [90]. They used four data acquisition modules to increase the imaging speed of the system significantly. A volumetric PA image of the entire breast was achieved within a single breath-hold (~15 s), thus minimizing motion-related artifacts in the images. In seven patients with malignant tumors, they quantified the vessel density of the tumors and compared it with the normal tissue (Figure 3c). The results clearly showed the differences between malignant and normal tissues, but the small number of samples limited statistical verification. In addition to the volumetric geometry, linear array transducers were also used for breast imaging. Wang et al. demonstrated the feasibility of PAI using a linear array transducer on a healthy volunteer [87]. From the same group, Nyayapathi et al. improved their system to achieve mammogram-like PA images by using linear array transducers located both superior and inferior to the breast [91]. They recruited 38 patients with malignant tumors, and acquired PA images of both diseased and healthy breasts with a wavelength of 1064 nm [92]. They defined four quantitative parameters which were calculated as the ratio of values in cancerous and normal tissues. (i) Vessel mean ratio: the ratio of the mean values of PA signals in the vessels. (ii) Contrast ratio: the ratio of the contrast that was calculated as the vessel values divided by the background signal. (iii) Standard deviation ratio of vessels: the ratio of the standard deviations of PA signals in the vessels. (iv) Standard deviation ratio of background: the ratio of the standard deviations of PA signals in the background. They used these to compare the PAI results of malignant and healthy breasts. The results showed statistical differences with *p*-values of ~0.05, but the diagnostic accuracy was not evaluated. Thus, the imaging results showed the feasibility of visualizing the vascular structure in human breasts, but multispectral analysis to assess the functional information was still missing.

Toi et al. investigated multispectral PA analysis using two wavelengths (755 and 795 nm) [93]. They recruited 22 patients with malignant tumors, and successfully visualized tumor-related blood vessels using a hemispherical array transducer (Figure 4a). They defined a novel imaging parameter named the S-factor, which depends upon the sO_2_ level and HbO composition in blood vessels. The comparison of vessels before and after chemotherapy showed the degradation of the S-factor, which indicates hypoxia. Although they could acquire high-quality images for a relatively large number of subjects, no statistical analysis was performed for differentiating malignant masses from benign or normal tissues. Diot et al. evaluated multispectral PAI with 28 wavelengths for the human breast in the range of 700–970 nm at 10 nm intervals [94]. Using the spectral unmixing of the multispectral data, they obtained the distributions of HbO, HbR, lipid, and water within the tissue. The corresponding total blood volume, which was related to the HbT level, was also calculated. They compared the findings between 10 patients with malignant tumors and 3 healthy volunteers. Compared to the healthy tissues, the malignant tumors showed significantly high values of total blood volume, indicating angiogenesis in the tumorous region. Statistical analysis was still unavailable due to the small number of patients. Thus, further statistical studies with a sufficient number of subjects are necessary to indicate the reliability of the results.

Neuschler et al. reported classification results using multispectral PA data on a large number of subjects (1079 benign and 678 malignant cases) [95]. They also presented a statistical analysis of their results. They used a laser with two wavelengths (757 and 1064 nm) to visualize HbO, HbR, and HbT distributions in human breasts. From the acquired images, the authors quantified five scores based on the corresponding features (Figure 4b). (i) Vessel score: the combination of the number of individually resolvable vessels and their relative degree of deoxygenation. (ii) Blush score: the average volume of vessels that are too small to distinguish. (iii) Hemoglobin score: the amount of hemoglobin. (iv) Boundary zone score: vascular morphology and deoxygenation in the boundary of nodules. (v) Peripheral zone score: the number of radiating vessels in the peripheral region. They combined the PA scores with the BI-RADS (breast imaging reporting and data system) grades, the conventional image-based risk stratification method for breast cancers. In benign cases, PA scores downgraded 34.5% of high-grade BI-RADS masses, while only 6.0% were upgraded from low BI-RADS grades. In contrast, PA scores upgraded 30.6% of malignant masses from low BI-RADS grades, and downgraded 16.5% of high-graded malignant masses. These results showed that the PA scores could correct the BI-RADS grades to improve diagnostic accuracy. The classification result yielded a sensitivity of 98.6% and a specificity of 43.0%. This is the largest clinical study with patients recruited from multiple sites, showing the clinical feasibility of multispectral PA analysis. The subsequent study validated the PA-based BI-RADS correction with 209 patients [96]. For 47.9% of benign masses, the proposed PA analysis method successfully downgraded the BI-RADS scores to two or three. The original scores of those masses were 4a or 4b, which are classified as highly suspicious for cancer in BI-RADS. The result showed the great potential of the PA-guided scoring method to decrease unnecessary biopsies or surgical operations for breast cancer patients.

The breast is one of the accessible organs which can be imaged using PAI. In addition to being superficial with protruding geometry, the homogeneous structure of the breast is optically transparent compared to other organs, lending itself as an ideal organ to be imaged using PAI. Thus, many clinical trials have been conducted to identify breasts cancer using multispectral PA images (Table 2). These studies have demonstrated the feasibility of PAI with a relatively large number of subjects. However, further validation is required for its successful translation to the clinic, as outlined here. (i) Improved specificity is needed for the reliability of this method. (ii) The study of additional excitation wavelengths and multiparametric analysis will allow the opportunity to define additional quantification methods, increasing the classification accuracy further. (iii) Imaging with contrast agents will assess the lymph node metastasis, which is one of the key factors for treatment planning [97,98,99].

## 5. Discussion and Outlook

Multispectral PAI is a promising method for investigating the metabolism of cancer cells and the molecular composition of biological tissue without injecting any contrast agents. In most cases, the quantification of sO_2_ level has been used for detecting tumor hypoxia, which is one of the well-known characteristics of cancerous tissue [100]. Using the functional information of PAI, clinical trials have been conducted for visualizing, diagnosing, and analyzing tissue in various clinical applications including cancers [36,101], melanoma [52,53], and vascular diseases [102].

This review summarized multispectral PAI for analyzing thyroid and breast cancers in humans. The multispectral PA analysis results showed the feasibility of this technique for classifying cancerous nodules from healthy tissues, but some limitations remain to be overcome for successful clinical translation. (i) The reproducibility of initial results should be verified and validated with larger-scale clinical studies. (ii) In addition to a simple cancer classification, various types of diseased nodules should be differentiable. (iii) A method to improve both sensitivity and specificity should be developed and verified. To solve the issues above, clinical PAI systems should be also improved with the following points: (i) real-time quantification for multispectral PA features, (ii) enhanced image quality, (iii) improved imaging depth, (iv) compact systems for better mobility, and (v) a user-friendly interface.

Although PAI is still in the initial stage of clinical translation, many studies have already established, with statistical analyses, the feasibility of distinguishing cancerous masses from benign tissues. With additional verification using a larger number of patients from multiple sites, a PAI-based modality could become an excellent method for screening cancer patients. In the near future, with sufficient data collected, machine learning techniques could be used to further increase classification accuracy.

## Figures and Tables

**Figure 1 metabolites-12-00382-f001:**
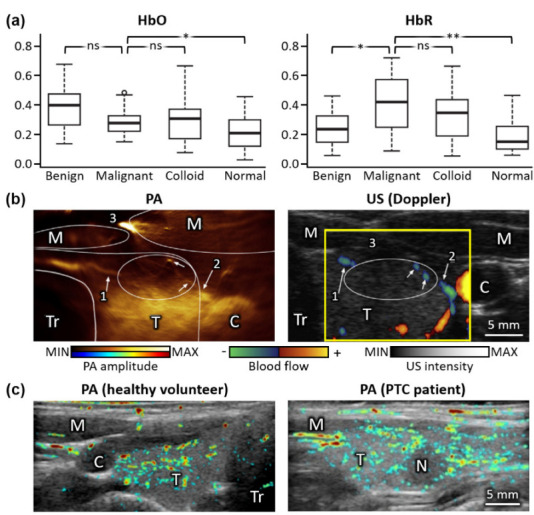
Initial results on PA analysis of thyroid nodules. (**a**) Student *t*-test results of HbO and HbR components in excised thyroid between malignant and nonmalignant (benign, colloid, and normal) tissues. **, *p* ≤ 0.01; *, *p* ≤ 0.05; ns, not significant. (**b**) PA (pseudo-colored), US (grayscale), and US Doppler (yellow box) images of thyroid from a healthy volunteer in vivo. The numbers indicate the locations of prominent blood vessels for comparison between PA and US Doppler images. (**c**) Overlaid PA and US images from healthy volunteers and PTC patients in vivo. PA, photoacoustic; US, ultrasound; HbO, oxy-hemoglobin; HbR, deoxy-hemoglobin; M, muscle; T, thyroid; Tr, trachea; C, carotid artery; N, nodule. The images are reproduced with permission from [71,72,73].

**Figure 2 metabolites-12-00382-f002:**
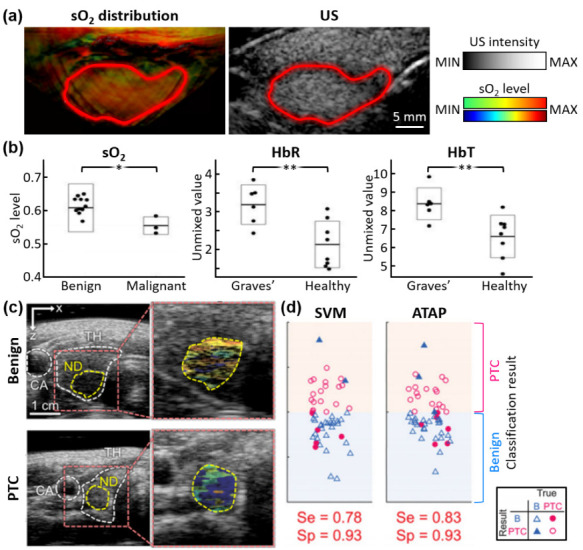
Multispectral PA analysis of thyroid nodules in vivo. (**a**) Spectrally unmixed sO_2_ distribution and corresponding US images of the thyroid region. (**b**) Student’s *t*-test results between diseased and non-diseased tissues, based on unmixed sO_2_, HbR, and HbT. (**c**) sO_2_ distributions in benign and PTC nodules overlayed on the corresponding US images. (**d**) Classification results using the SVM-trained three-dimensional decision function and the ATAP scoring method. PA, photoacoustic; US, ultrasound; sO_2_, hemoglobin oxygen saturation; HbR, deoxy-hemoglobin; HbT, total hemoglobin; PTC, papillary thyroid cancer; SVM, support vector machine; ATAP, the American Thyroid Association and the PA probability of PTC; Se, sensitivity; Sp, specificity; CA, carotid artery; TH, thyroid; ND, nodule; **, *p* ≤ 0.01; *, *p* ≤ 0.05. The images are reproduced with permission from [74,75].

**Figure 3 metabolites-12-00382-f003:**
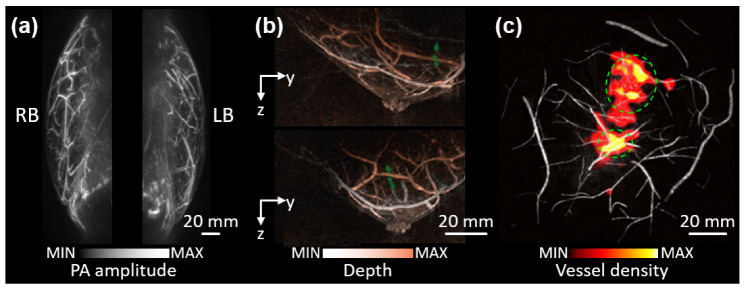
Volumetric PA images of human breasts in vivo. (**a**) Representative PA images of both breasts from a healthy volunteer. (**b**) Depth-resolved PA images of a healthy breast at different angles of view. (**c**) Vessel density map on the gray-scaled PA image. Green dashed circles denote the position of tumors. PA, photoacoustic; RB, right breast; LB, left breast. The images are reproduced with permission from [88,89,90].

**Figure 4 metabolites-12-00382-f004:**
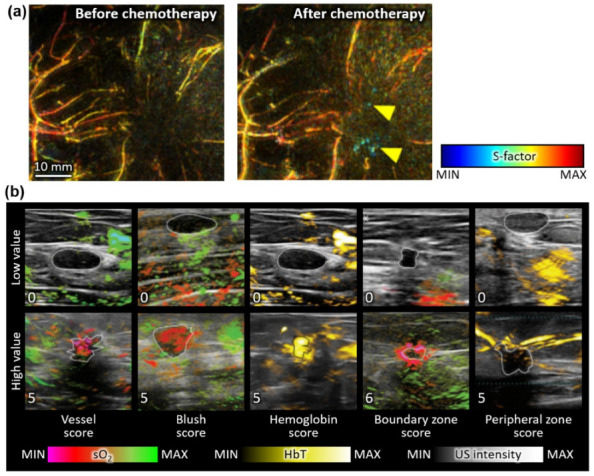
Multispectral PA analysis of human breasts in vivo. (**a**) The distribution of the S-factor before and after chemotherapy. The yellow marks show hypoxic signals in the tumor after therapy. (**b**) Representative multispectral PA images for minimum and maximum scores for each PA feature. PA, photoacoustic; US, ultrasound; sO_2_, hemoglobin oxygen saturation; HbT, total hemoglobin. The images are reproduced with permission from [93,95].

**Table 1 metabolites-12-00382-t001:** Summary of the system configurations and PA analysis results of the human thyroid cancers. PA, photoacoustic; US, ultrasound; PTC, papillary thyroid cancer; $f_{c}$, center frequency; λ, excitation wavelength; Se, sensitivity; Sp, specificity; n/a, not applicable.

Imaging Method	Imaging System	Transducer		Imaging Depth [mm]	Frame Rate [fps]	λ [nm]	# of Subjects	Accuracy [%]	Ref.
		Type	$f_{C}$ [MHz]						
ex vivo	n/a	Linear	5	5	10	760, 850, 930, 970	13 malignant 30 benign 13 colloid 32 normal	Se. 69.2 Sp. 96.9	[71]
in vivo	Terason 2000+, Teratech	Arc	7.5	~20	10	800	2 healthy	n/a	[72]
	Resona7, Mindray Bio-Medical Electronics	Linear	5.8	3.5	10	1064	10 PTC 3 normal	n/a	[73]
	MSOT Acuity Echo, iThera Medical	Arc	3	~20	25	700, 730, 760, 800, 850, 900, 920, 950	6 Graves’ disease 3 malignant 13 benign 8 healthy	n/a	[74]
	EC-12R, Alpinion Medical Systems	Linear	7.5	~30	5	700, 756, 796, 866, 900	23 PTC 29 benign	Se. 83 Sp. 93	[75]

**Table 2 metabolites-12-00382-t002:** Summary of the system configurations and PA analysis results of human breast cancers. PA, photoacoustic; US, ultrasound; $f_{c}$, center frequency; λ, excitation wavelength; Se, sensitivity; Sp, specificity; n/a, not applicable.

Imaging System	Transducer		Imaging Depth [mm]	Imaging Time [min]	λ [nm]	# of Subjects	Accuracy [%]	Ref.
	Type	$f_{C}$ [MHz]						
Custom	Hemi- spherical	2	53	3.2	756	4 healthy	n/a	[88]
Custom	Arc	1	22	4	755, 1064	2 healthy	n/a	[89]
SonixDAQ, Ultrasonix Medical	Ring	2.25	~40	~15 s	1064	7 malignant 1 healthy	n/a	[90]
Vintage 256, (Verasonics)	Linear	2.25	~70	1	1064	38 malignant	n/a	[92]
PAM-03, Canon-Optosonics	Hemi- spherical	2	27	4	755, 795	22 malignant	n/a	[93]
Custom	Arc	5	~20	2–4	700–970 (10 nm interval)	10 malignant 3 healthy	n/a	[94]
Imagio, Seno Medical Instruments	Linear	10	~30	Real-time	755, 1064	1079 benign 678 malignant	Se. 98.6 Sp. 43.0	[95]

## Data Availability

Not applicable.

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
