# Peer review of "Review on Multispectral Photoacoustic Analysis of Cancer: Thyroid and Breast"

_metabolites, 2022, doi:10.3390/metabo12050382_

Round 1

Reviewer 1 Report

This review summarizes the methods and results of clinical photoacoustic trials to date available in the literature to classify cancerous tissues, specifically the thyroid and breast. This topic is cutting-edge and photoacoustic imaging has big potential in clinical use. This review gives the readers a comprehensive understanding of PAI in clinical applications. I recommend the acceptance of this manuscript after minor changes.

  1. NIR-II photoacoustic imaging has superiorities over NIR-I photoacoustic imaging owing to higher penetration depth and lower MPE. Authors should talk about this in the revised manuscript.
  2. Contrast agents are usually necessary in conducting photoacoustic imaging. Some representative contrast agents including organic and inorganic ones should be discussed.
  3. The development of imaging instrument for clinical PAI should be introduced.

Reviewer 2 Report

Please explain why you limit your discussion of cancer diagnosis using multispectral PA to thyroid and breast cancer. Are other cancers not important for evaluating PA? Or has the application of PA not been studied in other cancers?

In the title of Fig. 1, (b) is missing.

In line 108, numbers for four different wavelengths are not listed.

Reviewer 3 Report

A nice review paper about photoacoustics on thyroid and breast malignancy. I believe the paper is nicely structured. I have some suggestions to improve the paper
Figure1b- What do the numbers represent on the images?
--In the discussion section, please add information about the limitations of the reviewed papers, and suggestions on how to address these limitations. I believe this would be very helpful for the readers. Discussion section should be expanded.  

Thank you for considering my recommendations. 

Reviewer 4 Report

The manuscript entitled “Review on Multispectral Photoacoustic Analysis of Cancer: Thyroid and Breast” by Seongyi Han, Haeni Lee, Chulhong Kim and Jeesu Kim provides an interesting overview over some examples of clinical trials on the integration of photoacoustic imaging in the diagnostic chain for critical conditions like thyroid and breast cancer. It places particular emphasis on the final performance of various systems and protocols in terms of sensitivity and specificity with respect to the state of the art. The subject matter is of much current interest and the point of view of the authors is original, as it elaborates more on the translational perspectives in specific case scenarios than the technological details that may be found in other recent reviews (10.1016/j.pacs.2019.100144; 10.3390/opt2010001; 10.1007/s13534-018-0062-7 and many more). The quality of the presentation is excellent as well. However, prior to publication, I recommend the following minor revision.

“The resulting PA images, formed from the acoustic wave, represent the optical absorption characteristics of the underlying biological tissue.” I understand that it is unfortunately much more complex than that, because the photoacoustic images return the product of local optical absorbance times local intensity of light times other factors that depend for instance on local pulse width after multiple scattering, plus pure acoustic reflections etc.. A word of caution should probably be added from the beginning.

“Four different wavelengths (c nm)…” What is the phrase in parentheses?

“… which are PA spectral gradient, relative sO2 level, skewed angle of sO2 distribution, were quantified and used for training decision function…” Please provide a brief description of these features.

“… (i.e., vessel mean ratio, contrast ratio, standard deviation ratio of vessels, and standard deviation ratio of background)…” or “… (vessel, blush, hemoglobin, boundary zone, and peripheral zone scores)…” Same as above. Without some definition, I believe that it is worthless to mention these names.

“… suspicious 4a or 4b BIRDS grades.” Please fix it.

“(3) Imaging with contrast agents will assess the lymph node metastasis, which is one of the key factors for treatment planning.” Please add refs to clinical and innovative contrast agents for photoacoustic imaging, such as 10.1021/ja412001e, 10.1002/adma.201805875, 10.3390/nano11010116, etc..

After these points will be addressed, in my opinion this manuscript will be ready for publication in MDPI Metabolites.

Round 2

Reviewer 2 Report

The manuscript has been correctly revised I recommend that this manuscript be accepted as is.

Author Response

Thank you for the very positive comment.

Reviewer 3 Report

Thank you for considering my suggestions. I only have a minor suggestion. Your response 'The numbers indicate the location of the surrounding blood vessel' is not clear to me. I would appreciate it if you could further clarify this point. Thank you very much. 

Author Response

Thank you for the very positive comment. We have modified the caption of Figure 1 as follows.

“The numbers indicate the locations of prominent blood vessels for comparison between PA and US Doppler images.”
